# Landslide forecasting and factors influencing predictability

**Emanuele Intrieri[1]\*, Giovanni Gigli[1]**

*[1] Department of Earth Sciences, University of Studies of Firenze, via La Pira 4, 50121 Firenze, Italy.*

*\*Corresponding author*

**ABSTRACT**

Forecasting a catastrophic collapse is a key element in landslide risk reduction, but also a very difficult task, owing to the scientific difficulties in predicting a complex natural event and also to the severe social repercussions caused by a false or a missed alarm. A prediction is always affected by a certain error, however when this error can imply evacuations or other severe consequences a high reliability in the forecast is, at least, desirable.

In order to increase the confidence of predictions, a new methodology is presented here. Differently from traditional approaches, it iteratively applies several forecasting methods based on displacement data and, also thanks to an innovative data representation, gives a valuation about the reliability of the prediction. This approach has been employed to back-analyse 15 landslide collapses. By introducing a predictability index, this study also contributes to the understanding of how geology and other factors influence the possibility to forecast a slope failure. The results showed how kinematics, and all the factors influencing it such as geomechanics, rainfall and other external agents, are the key concerning landslide predictability.

*Keywords: landslides; forecasting; geomechanics; early warning; time of failure; slope failure*

**INTRODUCTION**

Natural disaster forecasting for early warning purposes is a field of study that drew the media attention after events such as the 26[th] December 2004 tsunami of Sumatra. Predicting landslides, with respect to other natural hazards, is a complex task due to the influence of many factors like geomechanical properties, rainfall, ground saturation, topography, earthquakes and many others. So far, few empirical landslide forecasting methods exist (Azimi et al., 1988; Fukuzono, 1985a; Mufundirwa et al., 2010; Saito, 1969; Voight et al., 1988) and none furnishes a reliability degree about the prediction, making them unsuitable for decision making. In particular, when mentioning geomechanics, the reference is to the study of the behaviour of a landslide concerning its deformation with relation to the applied stress, with special reference to its post-rupture conditions.

In the present paper research an approach to perform probabilistic forecasting of landslide collapse is presented. This has been achieved by reiterating several predictions using more forecasting methods at the same time on multiple time series. This approach may have important applications to civil protection purposes as it provides the decision makers with a level of confidence about the prediction. Furthermore, this study, performed on 15 different case studies, shows how the possibility or not to forecast the time of collapse of a landslide is affected by geomechanical or geomorphological features as much as by circumstantial conditions.

**The inverse velocity forecasting method**

Forecasting activity can be considered the fulcrum of early warning systems (Intrieri et al., 2013), i.e. cost-effective tools for mitigating risks by moving the elements at risk away. For many natural phenomena forecasting is common practice (for example for hurricanes; Willoughby et al., 2007), while for others is, at present, impossible (earthquakes; Jordan et al., 2011). Landslides lie in between. Their prediction can be performed through rainfall thresholds (Baum and Godt, 2010), but a more reliable approach should make use of direct measures of potential instability, such as displacements (Lacasse and Nadim, 2010; Blikra, 2008). A first issue is that only a small percentage of landslides in the world is appropriately monitored, that often monitoring is carried out for short periods not encompassing the final pre-failure stages, or may have been carried out with a too low temporal frequency that does not permit to follow the displacement trend. This also causes an insufficient knowledge of the geomechanical processes

leading to failure (here meant as the collapse following a sudden acceleration, either a first
movement or a reactivation), which is another responsible for our deficiencies in predicting
landslides.
In spite of this, few empirical methods for predicting the time of failure based on movement
monitoring data have been developed (Azimi et al., 1988; Fukuzono, 1985a; Mufundirwa et al.,
2010; Saito, 1969) and further investigated on a physical basis (Voight et al., 1988). They are all
based on the hypothesis that if a landslide follows a peculiar time-dependant geomechanical
behaviour (called creep; Dusseault and Fordham, 1994), it will display a hyperbolic acceleration
of displacements before failure; by extrapolating this trend from a displacement time series
through empirical arguments, it is possible to obtain the predicted time of failure. However such
methods do not always produce good results. In fact, other than the limitation of working only
with creep behaviour, sometimes the tertiary creep can evolve such rapidly that a sufficient lead
time for evacuation is simply not possible (IEEIRP, 2015). In other cases natural or instrumental
noise can hamper the predictions and require post-processing to allow for effective warnings
(more details on the types and effects of noise can be found in Carlà et al., 2016). Other authors
also contributed to methodologies to exploit and optimize the classic forecasting methods (Crosta
and Agliardi, 2003; Dick et al., 2015; Manconi and Giordan, 2015).
One of the most famous methods is Fukuzono's (1985a), which derives from Saito's (1969),
from here on simply called F and S method, respectively. It requires that during the acceleration
typical of the final stage of the creep (tertiary creep), the inverse of displacement velocity ($v^{-1}$)
decreases with time. The collapse is forecasted to occur when the extrapolated line reaches the
abscissa axis (corresponding to a theoretical infinite velocity). Such line may either be convex,
straight or concave (Fukuzono, 1985a). When it is straight this phenomenon is sometimes
referred to as Saito effect (Petley et al., 2008).
The possibility to find landslides showing the Saito effect has been related to the mechanical
properties of the sliding mass. However there is no general consensus on this issue.
According to some authors (Petley, 2004; Petley et al., 2002), in order to display the Saito effect,
landslides need to display a brittle behaviour (which indicates a drop from peak strength to
residual strength value, deformation which is concentrated along a well defined shear surface,
sudden movements and catastrophic failure, usually associated with crack formation in strong
rocks); furthermore only brittle, intact rocks evolve in catastrophic landslides and therefore can
be predicted; for others (Rose and Hungr, 2007), on the opposite, landslides displaying the Saito
effect must have ductile failures in order to be forecasted (i.e. slower, indefinite deformation
along a shear zone and under a constant stress, typical of sliding on pre-existing surfaces of soft
rocks), as brittleness is characterized by sudden, impossible to anticipate, ruptures.
This complex subject is made even more difficult due to the influence of external factors
(rainfall, earthquakes, excavations), structural constraints (joints, faults, contacts with different
lithologies) and sometimes unknown elements within the mass (the conditions of the shear
surface, the history of the landslide, the presence of rock bridges). Therefore it is often hard to
establish the mechanical behaviour and even more to find an exact correlation between the
mechanical behaviour of a landslide and the possibility to predict its failure.
**The concept of predictability**
Before assessing the influence of geomechanics on the predictability of a landslide it is first
necessary to address the concept of predictability.
In literature (Azimi et al., 1988; Hutchinson, 2001; Mufundirwa et al., 2010; Rose and Hungr,
2007) there are papers that deal with "predictions" made in retrospect, that is thorough post-
event analyses showing the signs of a critical pre-collapse acceleration; however whether such
signs would have been unambiguous or would have granted a sufficient lead time is often
neglected.
On the other hand in this research an operational definition of predictability is considered
(integrating the one of early warning system; UNISDR, 2009) as the feature possessed by a
landslide which allows one to forecast its collapse with reasonable confidence and sufficiently in
advance, permitting the dispatch of meaningful warning information to enable individuals,
communities and organizations threatened by the hazard to prepare and to act appropriately and
in sufficient time to reduce the possibility of harm or loss. Therefore, displaying the Saito effect
is not the only prerequisite for an operational prediction, there is also the need for repeated time
of failure forecasts fluctuating around a constant time value placed not too close in the future.
This has been achieved through the reiterative approach and the graphical representation
described in the following paragraph. Finally a semi-quantitative parameter called Prediction
Index is defined in order to address the success of the predictions.
**METHODS**
The usual way to apply landslide forecasting methods based on displacements, is to obtain a
single predicted time of failure ($t_f$) and to update such prediction as soon as new data are
gathered (Rose and Hungr, 2007). This is a deterministic approach, since the real time of failure
($T_f$) is predicted through a single inference. Even if sometimes more predictions are made
together with new data, usually only one (the most recent) is used.
On the other hand, in order to account for the uncertainty of the methods and complexity of the
phenomena, predictions should have a certain confidence. Confidence may be quantitatively
assessed by using the standard deviation of the forecasts $t_f$ as a proxy. In fact the standard
deviation furnishes the dispersion (i.e. the precision) of the predictions, which may be used to
calculate a time window within which the collapse is more likely to occur. Therefore the lower
the standard deviation of a set of forecasts, the higher would be their reliability and the
confidence. This is especially important for operative early warning systems. This probabilistic
approach is achieved by reiterating the equations from Saito (1969), Fukuzono (1985a) and
Mufundirwa et al. (2010) (the latter method will be called M method from here on) for finding $t_f$,
using continuously new data and enabling the calculation of the standard deviation.
The predictions are plotted versus the time when they have been made (time of prediction, $t_p$).
We call these diagrams prediction plots (Figure 1). A prediction is considered reliable when the
inferences oscillate around the same $t_f$. Figure 1 also shows that since reliable predictions usually
display an oscillatory trend, the most updated one is not necessarily the most accurate, contrarily
to what is usually believed (Rose and Hungr, 2007) in fact, the length of the dataset is more
important, from which $T_f$ can be estimated through simple statistical analyses (like mean and
standard deviation).
Since in some cases a single forecasting method can fail to give satisfactory results, in order to
improve even more the confidence in the predictions, a multi-model approach is adopted together
with the probabilistic approach. In fact, according to the Diversity Prediction Theorem (Page,
2007; Hong and Page, 2008), diversity in predictive models reduces collective error. The highest
confidence, of course, is reached when all the employed method independently converge towards
the same result.
On the other hand, confidence it may also be considered as a qualitative increase in the
awareness of the decision makers that can estimate the time of failure of a landslide by
evaluating a large set of different predictions and their dispersions.

For this research the results from S and F methods have been confronted and from the method by Mufundirwa et al. (2010). The equations used for the iteration are obtained from the respective authors and are:

$$t_r = \frac{t_2^2 - (t_1 \cdot t_3)}{2t_2 - (t_1 + t_3)}, \quad (1)$$

for S method, where $t_1$, $t_2$, $t_3$ are times taken so that the displacement occurred between $t_1$ and $t_2$ is the same as between $t_2$ and $t_3$.

$$t_r = \frac{t_2 \frac{1}{v_1} - t_1 \frac{1}{v_2}}{\frac{1}{v_1} - \frac{1}{v_2}}, \quad (2)$$

for F method, where $v_1$ and $v_2$ are the velocities at arbitrary times $t_1$ and $t_2$.

$$t\frac{dD}{dt} = t_r\frac{dD}{dt} - B, \quad (3)$$

for M method, where $D$ is the displacement and $t_r$ is the angular coefficient of the line represented in a $t\frac{dD}{dt} = f\left(\frac{dD}{dt}\right)$ space having $B$ as the intercept. For the purposes of this paper $t_r$ expressed in all these equations is equivalent to $t_f$.

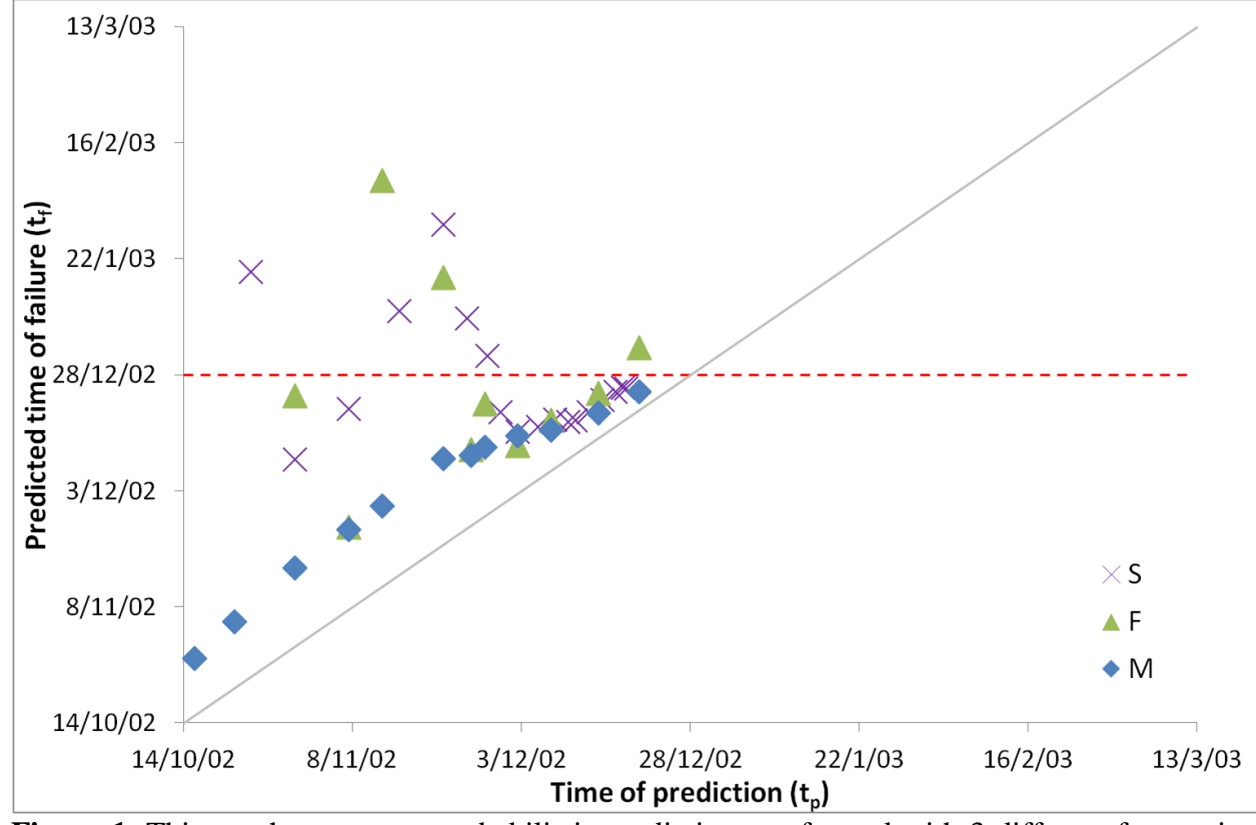

**Figure 1**. This graph represents probabilistic predictions performed with 3 different forecasting methods (Fukuzono, 1985a; Mufundirwa et al., 2010; Saito, 1969) applied to the MB34-35'

displacement time series of Mount Beni landslide (Gigli et al., 2011). The horizontal dashed line
indicates the observed time of failure ($T_f$) and the grey diagonal line the equality between $t_f$ and
$t_p$. Therefore the vertical distance between a point and the dashed line indicates the prediction
error. The vertical distance between the diagonal line and a prediction above it is the life
expectancy of the landslide at the time of prediction. In this case the predictions obtained through
S and F methods give a good estimation of $T_f$, while the one from Mufundirwa et al. (2010)
consistently forecasts the collapse few days ahead.
The proposed procedure consists in iteratively calculating the time of failure $t_f$ by using the
aforementioned methods and to repeat the calculation as soon as new monitoring data are
available. All the forecasts are recorded together with the time when they are made, in order to
create a time series of $t_f = f(t)$. This can be represented in a prediction plot having $t_f$ and $t$ (the
time when the prediction is made) as coordinates. Finally, from the distribution of the forecasts
with time it is possible to assess the time of failure.
**TIME OF FAILURE PREDICTION**
In order to find a relation between the predictability of a failure and the geological features of the
landslide, S, F and M methods have been applied to a number of different real case studies. Some
geological features of interest relative to such cases are reported in TABLE 1, when they were
known or applicable. Concerning brittleness, since it was rarely explicitly stated in the
referenced articles, it was assessed based on information such as the type of material, the
presence of a reactivated landslide, the weathering and the shape of the displacement time series.
Since this lead to approximations, brittleness has been evaluated using broad and qualitative
definitions.
Since $T_f$ must be known in order to assess the quality of predictions, all the case studies are from
past landslides that have already failed. Therefore the respective time of failures are all a
posteriori known.
A few representative examples of prediction plots are showed in Figure 1 and Figure 2. Mount
Beni landslide is a 500.000 m$^3$ topple that evolved as a rockslide (Gigli et al., 2011). It developed
on a slope object of quarrying activity. The predictions oscillate quite regularly around the
observed time of failure ($T_f$, dashed line in Figure 2). It is this convergence that permits to
correctly forecast the collapse a priori at least since late November, i.e. a month before the
failure, whereas a single forecast would not be able to give a confidence of the prediction. The
three methods are similar to the point that S and F previsions can be partially overlapped. M
previsions overlap as well but only in the final part. The M method alone would not be sufficient
for spreading a reliable alarm as the single forecasts do not converge but move forward to a
different time of failure as the time passes by.
Similar behaviours can be observed also for the cases of Figure 2 that display landslides with a
different array of geological features (as seen in TABLE 1). The best results are obtained when
the forecasts oscillate around $T_f$ with sufficient time in advance (as for Vajont and, limited to F
method, for Liberty Pit) or when they consistently give the similar $t_f$ (as for the artificial
landslide E, where the terms "artificial landslide" indicate a landslide recreated in laboratory
with an artificial slope). In other cases (Avran valley and, limited to S and M method, for Liberty
Pit) the predictions are too scattered or simply never converge toward a single result, thus
making it impossible to foresee a reliable time of failure.
Notably, considering for example only the results of the S method in the case of the Avran valley
landslide, since the end of September the forecasts are constantly furnishing a time of failure
preceding the actual $T_f$. Although this may be considered a case of safe predictions (that is an
error not producing a false positive and therefore not dangerous for the elements at risk), this
also means that, at every forecast that is made, $t_f$ is postponed. Given a series of ever increasing
values of $t_f$, it is impossible to assess which of them (if any) can be assumed as a good estimate
of the actual time of failure. However, since the time series of predictions is long enough, past
forecasts (before early September) furnish values of $t_f$ that, if considered together with the late
ones, centre the value of $T_f$. Therefore it is clear how a prediction plot may allow decision
makers to make more aware evaluations of the time of collapse of a landslide.
The results of the prediction plots can be roughly summarized reporting the mean and standard
deviation of the forecasts for each method (Figure 3).

## TABLE 1. LANDSLIDE CASE HISTORIES

| Name | Material | Type | Brittleness | Volume (m$^3$) | Predisposing factor | Trigger | History | Basal geometry | Ref. * |
|---|---|---|---|---|---|---|---|---|---|
| Liberty Pit | Weathered quartz monzonite | Rockslide? | Medium/high | 6x10$^6$ | N.D. | Blasts, pore water pressure | First time failure | Planar? | 1, 2 |
| Landslide in mine | Consolidated alluvial sediments, weathered bedrock | Deep-seated toppling in bedrock | Medium | 10$^6$ | Blasts, pore water pressure | N.D. | First time failure? | N.D. | 1 |
| Betze-Post | Weathered granodiorite | Rockslide? | Medium/high | 2x10$^6$ | N.D. | Rainfall | First time failure? | Wedge intersections? | 1 |
| Vajont | limestone and clay | Rock slide | High | 2.7x10$^8$ | N.D. | Pore water pressure | Reactivated | Concave | 1, 3 |
| Stromboli † | Shoshonitic basalts | Bulging (not a landslide) | Medium/high | N.D. | N.D. | Sill intrusion | First time failure | N.D. | 4 |
| Monte Beni | Ophiolitic breccias | Topple/rock slide | High | 5x10$^5$ | Rainfall, structure, basal excavation | N.D. | First time failure | Stepped | 5 |
| Cerzeto | Weathered metamorphic rocks on top, cataclastic zone and Pliocene clays | Debris slide-earth flow | Medium/low | 5x10$^6$ | Tectonized area, permeability differences | Prolonged rainfalls | Reactivated ? | Compound (steeper and irregular in the upper zone and gentler in the clays) | 6 |
| Rock mass failure Japan | Clayey limestone | Rockslide? | High (within limestone)? | 5x10$^2$ | "Structural complexity" (?) | Intense rainfall | First time failure? | Planar? | 7 |
| Asamushi | Liparitic tuff, jointed and weathered. Clay in the joints | | Medium/low | 10$^5$ | N.D. | N.D. | N.D. | Concave? | 7, 8 |
| Avran valley | Chalk | Rockslide | Medium/low | 8x10$^4$ | N.D. | N.D. | First time failure? | Convex | 9 |
| Giau Pass | Morainic material | Complex slide | Medium/low | 5x10$^5$ | N.D. | Pore water pressure | Preexisting shear surface | Composite | 10, 11 |
| Artificial landslide A | Loam | Earth slide | Low | N.D. | N.D. | Prolonged rainfall | First time failure | Planar | 12 |

| Artificial landslide B | Sand | Earth slide | Low | N.D. | N.D. | Prolonged rainfall | First time failure | Planar | 12 |
|---|---|---|---|---|---|---|---|---|---|
| Artificial landslide C | Sand | Earth slide | Low | N.D. | N.D. | Prolonged rainfall | First time failure | Convex | 12 |
| Artificial landslide D | Sand | Earth slide | Low | N.D. | N.D. | Prolonged rainfall | First time failure | Planar | 12 |

*The references used are numbered as follows: 1: Rose and Hungr, 2007; 2: Zavodni and Broadbent, 1980; 3: Semenza and Melidoro, 1992; 4: Casagli et al., 2009; 5: Gigli et al., 2011; 6: Iovine et al., 2006; 7: Mufundirwa et al., 2010; 8: Saito, 1969; 9: Azimi et al., 1988; 10: Petley et al., 2002; 11: Angeli et al., 1989; 12: Fukuzono, 1985b.

† The case of Stromboli is not relative to a landslide, rather to a volcanic bulging preceding a vent opening that was forecasted in a similar fashion of a landslide and therefore here included.



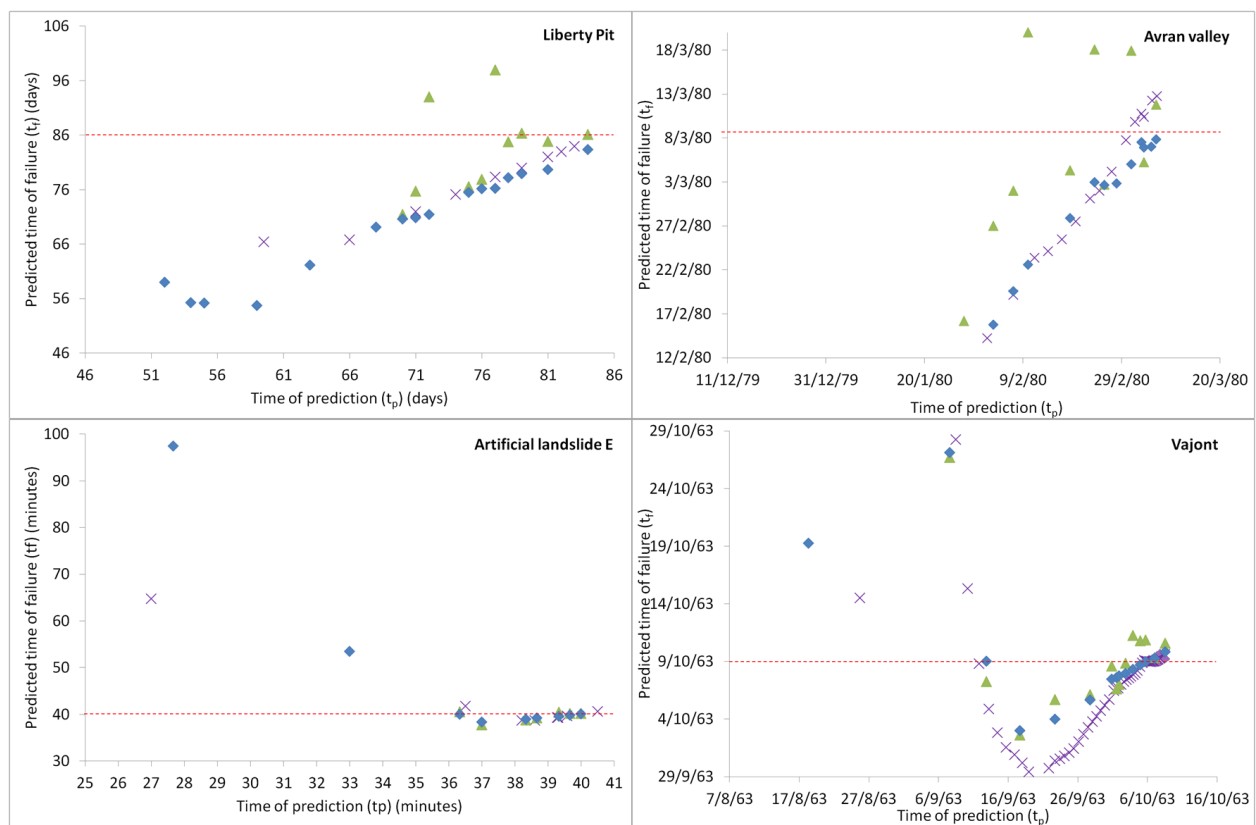

**Figure 2.** Prediction plots of four different case studies. The dashed line indicates $T_f$. The crosses
represent forecasts performed with S method, the triangles with F method and the diamonds with
M method. Note that F forecasts for Avran valley landslide include other less accurate values not
showed in the graph as they are out of scale.

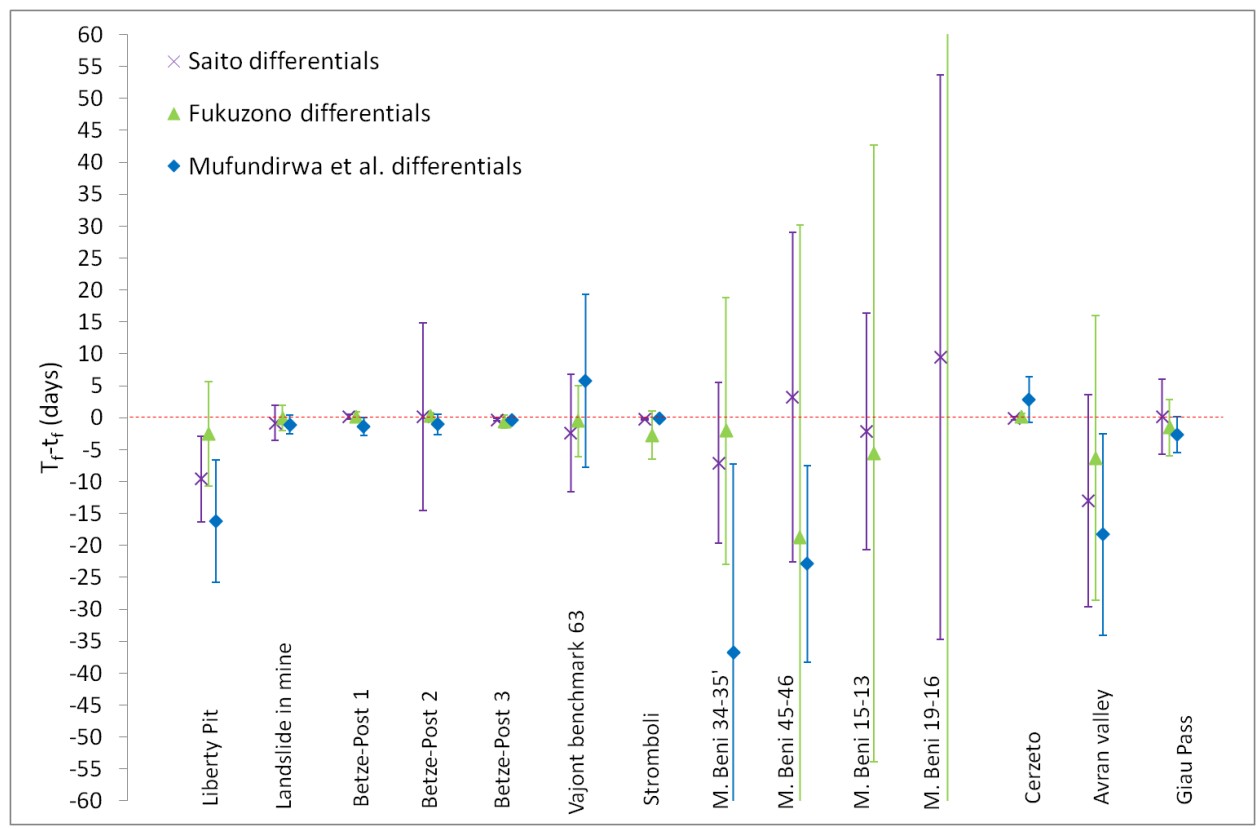

**Figure 3.** This graph represents for each method the differential between the mean of the
forecasts ($\bar{t}_f$) and the actual time of failure ($T_f$). Negative values are safe predictions as anticipate
the time of failure. The dashed line represents exact predictions ($T_f - \bar{t}_f = 0$). The standard
deviations of the forecasts are represented as error bars. For Betze-Post and Mount Beni
landslides, time series from different measuring points are reported. The rock mass failure,
Asamushi landslide and the artificial landslides are not shown as were monitored in a different
time scale (hours or minutes).
**PREDICTABILITY INDEX**
In order to evaluate the performance of S, F and M methods and to relate it to the characteristics
of the reported examples, an arbitrary scoring system has been implemented and attributed to
each prediction plot (considering that every time series has a prediction plot for each forecasting
method and that for some case studies more than one time series was available). This permits to
quantify the predictability of a collapse based on the prediction plot. A score from 1 to 5 has
been assigned according to the following criteria:
• 1 point: the prediction plot never converges on a single $t_f$ (typically $t_f$ increases at every
new datum available).
• 2 points: the predictions vary considerably at every new iteration. An average time of
failure ($\bar{t}_f$) can be extracted but with high uncertainty.
• 3 points: the predictions oscillate around $T_f$, although with a certain variance.
• 4 points: the predictions have a low variance although $\bar{t}_f$ is slightly different than $T_f$. Note
that when the variance was low, $\bar{t}_f$ and $T_f$ never differed greatly.
• 5 points: the prediction plot is clearly centred on $T_f$ therefore the reliability of $\bar{t}_f$ is high.
By summing the scores obtained from S, F and M prediction for each time series, what we call
the Predictability Index (*PI*) is obtained (TABLE 2). Since PI is a means to evaluate the overall
quality of a set of predictions (it requires to observe the time series of $t_f$ and confront it with $T_f$, it
is the predictability index) and also to compare the performance of different forecasting methods
with different case studies, naturally it can only be estimated after the collapse.
By using 3 forecasting methods, *PI* ranges from 3 (impossible to predict the time of failure) to 15
(the time of failure can be predicted in advance and with a high reliability). Though a certain
degree of subjectivity is unavoidable when assigning the scores, what matters here is the relative
difference of *PI* between the case studies. In such a way it is possible to understand in which
conditions a landslide is more or less predictable.

## TABLE 2. PREDICTABILITY INDEX

| Name | S | F | M | *PI* | Inverse velocity trend | Notes |
|---|---|---|---|---|---|---|
| Liberty Pit | 1 | 5 | 1 | 7 | Asymptotic (linear at the end) | Open pit mine, structural control of 2 intersecting faults |
| Landslide in mine | 5 | 5 | 5 | 15 | Linear | Open pit mine |
| Betze-Post 1 | 3 | 3 | 1 | 7 | Linear | Open pit mine |
| Betze-Post 2 | 4 | 5 | 4 | 13 | Linear | Open pit mine |
| Betze-Post 3 | 5 | 4 | 1 | 10 | Linear | Open pit mine |
| Vajont benchmark 63 | 5 | 5 | 5 | 15 | Linear | Air pressure and cementation caused catastrophic collapse |
| Stromboli | 1 | 2 | 2 | 5 | Asymptotic | Volcanic context |
| Mount Beni 12-9 | 4 | 5 | 1 | 10 | Concave | Back fracture |
| Mount Beni a'b' | 1 | 3 | 1 | 5 | Linear | Short time series |
| Mount Beni 15-13 | 5 | 3 | 1 | 9 | Linear | Internal fracture |
| Mount Beni 34-35' | 5 | 3 | 1 | 9 | Linear | Lateral fracture, short time series |
| Mount Beni 45-47 | 2 | 3 | 1 | 6 | Linear | Back fracture, short time series |
| Mount Beni 3-2 | 5 | 2 | 1 | 8 | Concave | Back fracture |
| Mount Beni 4'-6 | 1 | 4 | 1 | 6 | Linear | Back fracture, short time series |
| Mount Beni 24-23 | 4 | 2 | 1 | 7 | Linear | lateral fracture |
| Mount Beni 49-24 | 5 | 1 | 1 | 7 | Linear | Lateral fracture, short time series |
| Mount Beni 35'-36 | 2 | 5 | 1 | 8 | Linear | Lateral fracture, short time series |
| Mount Beni 33-35' | 3 | 3 | 1 | 7 | Linear | Lateral fracture, short time series |
| Mount Beni 36-37 | 4 | 3 | 1 | 8 | Linear | Lateral fracture |
| Mount Beni 19-16 | 2 | 2 | 1 | 5 | Linear | Lateral fracture |
| Mount Beni 19-17 | 1 | 2 | 1 | 4 | Linear | Lateral fracture, short time series |
| Mount Beni 33-34 | 4 | 2 | 1 | 7 | Linear | Internal fracture |
| Mount Beni 43-44 | 3 | 2 | 1 | 6 | Asymptotic (constant velocity at the end) | Internal fracture, short time series |
| Mount Beni 40-41 | 3 | 2 | 1 | 6 | Asymptotic (constant velocity at the end) | Internal fracture, short time series |
| Mount Beni 40-42 | 3 | 3 | 1 | 7 | Linear | Internal fracture, short time series |
| Mount Beni 45-46 | 3 | 2 | 2 | 7 | Linear | Back fracture, short time series |
| Mount Beni 1-2 | 4 | 2 | 1 | 7 | Linear | Back fracture |
| Cerzeto | 5 | 5 | 1 | 11 | Linear | N.A. |

| | | | | | | |
|---|---|---|---|---|---|---|
| Rock mass failure Japan | 2 | 2 | 1 | 5 | Convex | Open pit mine, very small landslide |
| Asamushi | 5 | 3 | 1 | 9 | Linear | N.A. |
| Avran valley 5 | 1 | 2 | 1 | 4 | Concave | N.A. |
| Avran valley 6 | 1 | 1 | 1 | 3 | Asymptotic | N.A. |
| Avran valley 7 | 1 | 2 | 1 | 4 | Concave | N.A. |
| Giau Pass | 3 | 3 | 1 | 7 | Asymptotic /concave | N.A. |
| Artificial landslide A | 5 | 5 | 5 | 15 | Convex | 40° artificial slope |
| Artificial landslide B | 2 | 2 | 3 | 7 | Concave | 40° artificial slope |
| Artificial landslide C | 1 | 2 | 3 | 6 | Linear (slightly convex) | 40° artificial slope |
| Artificial landslide D | 5 | 5 | 5 | 15 | Linear | 30° artificial slope |


**DISCUSSION**
TABLE 2 shows how the most predictable events ($PI > 8$) can display very different features and
are quite irrespective of the shape of the inverse velocity plot, the volume, the brittleness of the
material, the history of the landslide and so on (see also TABLE 1).
A comparison between Figure 3 and TABLE 2 illustrates how the mean and standard deviation
of the forecasts alone are not enough to represent the quality of predictions and, consequently,
the predictability of a landslide. In fact the importance of a single forecast strongly depends on
the time when it is made; for example, given the same set of forecasts ($t_{f,i}$), a higher $PI$ is
obtained if the first predictions done are the farthest from $T_f$ while the final ones tend to converge
to it; in this way the prediction plot assumes an oscillatory shape (as for S and F forecasts in
Figure 1). Conversely, if the same forecasts are made with a different order so that they get
closer and closer to $T_f$ as time passes by (that is $|t_{f,i} - T_f| < | t_{f,i-1} - T_f|$), then there is no $t_{f,i}$
prevailing on the others and it is not possible to define a more probable time of collapse (as for
M forecasts in Figure 1). However the average and standard deviation of $t_f$ are the same for both
cases and this explains why these two statistics alone are not as informative as a prediction plot.
From TABLE 2 it is also possible to assess which method gives the best results. The sum of the
scores for S, F and M is 119, 115 and 63 respectively. Overall S and F perform similarly, but for
a specific case study their effectiveness can be very different, therefore their result are
independent and not redundant; there is no indisputable clue suggesting when F method is more
performing than S and vice versa; nonetheless it appears that S is negatively influenced when the
displacement curve is not regularly accelerating (Liberty Pit, Stromboli), whereas for F a few
aligned points in the final tract in the inverse velocity plot are sufficient for predicting the failure;
however F forecasts are more disturbed when displacement data are noisy, since they use their
derivative (velocity) as input. Eventually M forecasts generally perform more poorly and rarely
(i.e. artificial landslides B and C) surpass those obtained from S and F methods.
Interestingly, different displacement time series belonging to the same landslide can display
different behaviours. This is a strong evidence that, even though the geological features do
influence the predictability of a landslide, assuming that they keep the same for the whole
landslide, other factors must determine the quality of the predictions. The last column of TABLE
2 shows for each time series what such factors could be, such as lithology (the asymptotic trends
of the cases of Avran valley and Giau Pass can be explained as consequences of a lowly brittle
material according to Petley's experiments; Petley, 2004), external forces (excavation in open pit
mines, volcanic activity, rainfall), local effects (structural constraints, displacement measured
relative to internal or lateral fractures not representing the general instability of the landslide),
quality of data (length of the time series, frequency of the observations, level of noise,
representativeness of the monitored point) etc.
All these case histories show that the main responsible for the predictability of a landslide, and
secondary also for the presence or not of the "Saito effect", is in a way or another connected to
geology. However this relation is not simple nor direct. Instead both the predictability and the
"Saito effect" depend on the kinematics of the landslide, since only a landslide accelerating with
a certain trend can be forecasted using S, F and M methods. Naturally, the kinematics in turn
depend on the geological conditions. In the complex relation between geology and kinematics
the aforementioned factors may intervene. Although their interaction may not be known, its
effect on displacement data can be easily measured. As a result it has been found that asymptotic
trends in the inverse velocity plot have been encountered also for first failure ruptures (as found
in some time series of Mount Beni landslide), contrarily to what is described by Petley (2004).
This can be explained as an effect of those interactions which may alter in an unknown way the
normal relation between geology and kinematics, thus making focusing on kinematics as the key
more reliable than relying on geology alone.
In fact, even though geomechanics is unquestionably a key factor, a complete geomechanical
characterization is often difficult to accomplish, especially in emergency situations. Hints of a
particular geomechanical behaviour are often derived from displacement data. Like in a black
box model, even if the real properties of a phenomenon are not known, conclusions may be
drawn from the output of those properties (i.e. the kinematics). In this case, importance has been
done to kinematics because what is generally measured by monitoring are displacement data.
Furthermore, many other unknown factors (rainfall, ground saturation, earthquakes, anthropic
disturbance etc.) are included in the black box model together with the geomechanics; this makes
it virtually impossible to know in advance what may be the degree of influence of geomechanics
alone with respect to other factors, thus leading to focusing on kinematics instead. Moreover,
even though geomechanics is a key element in determining landslide predictability (for example
because it is responsible for the creep behaviour), the results of the present study showed that
landslide prediction can be carried out with a variety of different geomechanical settings, as can
also be observed by comparing TABLE 1 (which furnishes evaluations concerning the
geomechanical properties of the case studies) with TABLE 2 (which states whether a collapse
was predictable or not).
The prediction plots clearly show that, contrarily to what is generally believed (Rose and Hungr,
2007), the last forecasts are not necessarily the most accurate and that past ones (starting from
the initiation of the tertiary creep) are essential to estimate the correct time of failure. In fact
older forecasts can be more accurate and in any case furnish precious information about the
general reliability of the final prediction, as explained above. Therefore the present study
highlights the importance of considering the whole set of predictions made with time. The
integration of more forecasting methods further raises reliability of the predictions, which is of
great importance for early warning systems, in particular when evacuations are envisaged.
Limitations of the proposed approach are those related to the intrinsic limitations of the
forecasting methods that have been integrated. In fact, since S, F and M methods are all based on
the creep theory, the occurrence of a tertiary creep phase slow enough to allow to monitor and
take action is necessary. Voight (1988) also assumes that there must be no external force acting
on the landslide, but the examples shown in this paper demonstrate that this may not represent a
limitation.
Figure 3 shows that the mean of the predictions can be used as a proxy for the time of failure but,
as stated above in this paragraph, it is also shown that the obtained accuracy may not be enough
as the mean does not exploits all the information provided by a prediction plot. Other statistical
indicators have been attempted but none of them appeared to better approximate the value of $T_f$,
mainly due to the difficulty of accounting for the important time factor in the forecasts and also
because not every prediction plot displays the characteristic oscillations. Therefore, the
interpretation of the prediction plot (and in particular of the dispersion of the forecasts with time)
represents the most valuable tool for decision makers, who, in this way, can make aware
judgements informed with a large set of quantitative and redundant data and therefore assessing
the "weight" of a single prediction by comparing it with many others.
Resuming, the proposed methodology can be summarized as in Figure 4.

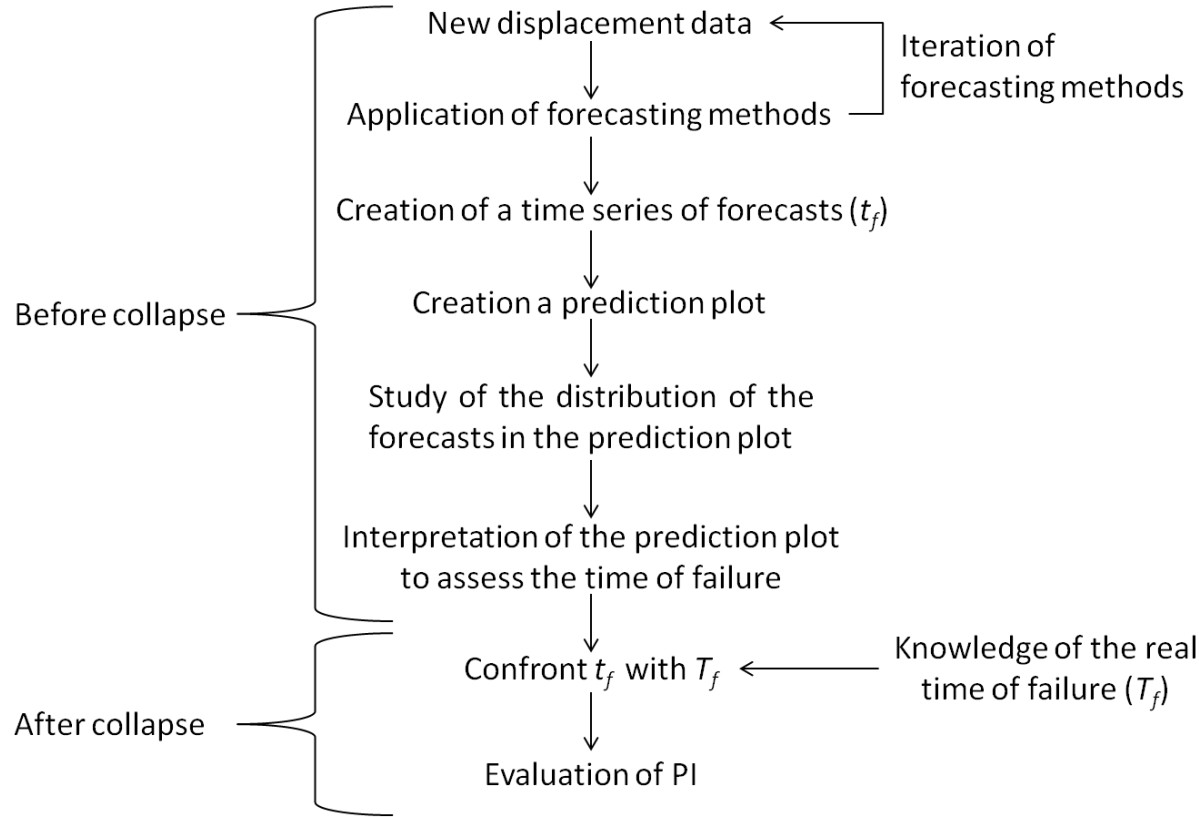

**Figure 4.** Flow-chart that synthesises the proposed procedure.
**CONCLUSIONS**
In conclusion, the main aspect of the proposed methodology concerns a way to produce and
represent forecasting data. Then this methodology is used to assess the influence of different
factors in the predictability of a landslide. The main results of such study are the following:
• Prediction plots are introduced as graphs showing the evolution of collapse forecasts with
time. Such plots provide more information than simple average and standard deviation of
the forecasts and improve the reliability of the final prediction.
• A predictability index (*PI*) has been introduced as a scoring system based on the
description of the prediction plot, in order to evaluate the quality of a set of predictions.
• The predictability of a landslide depends firstly on its kinematics and then on what
determines it (geology, external forces, local effects etc.).
• Landslide collapses can be forecasted whether they are in highly or lowly brittle
materials, in rock or in earth material, of different types, with different sliding surface
geometries, volumes and triggers.
• Contrarily to what is generally assumed (Voight, 1988; Rose and Hungr, 2007),
landslides can be forecasted also with external forces acting.
• The asymptotic behaviour of the inverse velocity curve does not imply that the landslide
cannot be correctly forecasted, even though it can hinder the prediction.
• The asymptotic behaviour may be induced by external factors, lithology and local effects,
rather than only by crack propagation. In fact asymptotic trends have been found in first
time failures and in both brittle and lowly brittle materials. The crack propagation
explanation is not neglected, but it may not represent the general rule.
• Most recent displacement monitoring data increase the confidence when estimating the
time of failure but do not necessary provide more accurate predictions than the older ones
(provided that they start from after the initiation of the tertiary creep).
• The developed approach integrates more forecasting methods to further improve the
reliability of the prediction.

**AUTHOR CONTRIBUTION**
E. Intrieri developed the idea and performed the analyses. G. Gigli supervised and improved the
manuscript.

**ACKNOWLEDGEMENTS**
The authors are thankful to Antonio Intrieri for his important technical contribution when
computing the calculations needed for this work.
No competing financial interests exist.

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
