# Peer review of "Landslide forecasting and factors influencing predictability Emanuele Intrieri1\*, Giovanni Gigli1 Department of Earth Sciences, University of Studies of Firenze, via La Pira 4, 50121 Firenze,"

_Natural Hazards and Earth System Sciences, 2016_

## Referee Comment (RC1) · Anonymous Referee #1 · 8 Aug 2016

Anonymous Referee #1

I appreciate the effort by the authors on pursuing a landslide prediction tool that accounts for the reliability in its predictions. The proposed methodology is based on careful consideration of the work done by others and supported by its implementation on several case studies. This is important work that should be encouraged in landslide research for risk management purposes.

I do have some general comments and discussion.

The authors state the importance of kinematics over geomechanics, based on their interpretation of results. I would suggest that not only does geomechanics play a major role in the kinematics of some of their case studies, but also that predictability of other landslide types not included in the database in this paper are likely controlled by the geomechanics. Clear examples are landslides in sensitive clays and other materials

prone to collapse.

The authors should also discuss the issue of timely predictability. Methods used to predict landslides that are based on displacement monitoring assume that slope collapse will be preceded by accelerations, sufficiently in advance to make adequate predictions followed by emergency measures. Again, landslides in sensitive clays and other collapsible materials are examples where this assumption might not be valid. Moreover, the recent failure of the Mount Polley Dam (IEEIRP, 2015) suggest that, under certain conditions, undrained responses leading to failure might not provide enough warning time for emergency plans to be in place. It is suggested the authors state such limitations of the methods proposed.

The methodology presented addresses the variability of the forecasting methods used. The reliability index, based on this variability, the convergence and non convergence of forecasts; appears to be a measure of data scatter and trend variation, rooted in the behavioural nature of the landslide in its pre-failure stage. To assess the reliability of any forecasting method, the range of forecasts for a number of case studies needs to be compared against observed time of failure. This requires, in my opinion, to subdivide the case dataset in groups of same landslide type, kinematics, materials, triggers, etc., and compare the forecasts with the observed times of failure.

For particular comments:

1.- How was brittleness assigned for the cases in Table 1?

2.- In Table 1, the event at Vaiont is classified as a "Rock Avalanche". This term refers to the material (rock) and its post-failure behaviour. I suggest it should be classified following its detachment process, as this is what we are monitoring prior to failure and would give more insight into the role of landslide kinematics vs. predictability.

3.- What are the artificial landslides?

For editorial comments:

1.- I suggest the improvement of the excel figures. fonts are too small, and layout is not technical. The text refers to dashed black and grey lines that appear continuous red and blue in the figures.

2.- Should the title read "...influence of geology on predictability" rather than "...influence of geology to predictability"?

References:

Independent Expert Engineering Investigation and Review Panel (IEEIRP) (2015) Report on Mount Polley Tailings Storage Facility Breach. Province of British Columbia. https://www.mountpolleyreviewpanel.ca

––––––––––––––––––––––––––––––

---

## Referee Comment (RC2) · Anonymous Referee #2 · 10 Aug 2016

Dear Editor, Please find here below my review of the paper nhess-2016-221:

Operative and reliable landslide forecasting and influence of geology to predictability By Emanuele Intrieri, Giovanni Gigli

This paper is related to new ways of forecasting the time of failure of landslides. It is based on the displacements interpretation by three time to failure existing approaches. The used of the variability of the three methods is proposed to assess the time of failure. The method is applied to several case study. In addition, more general consideration are made about the processes involved.

General comments

The method presented is innovative and interesting, but it seems that too much conclusions are from this research. First the title, is probably to pretentious, I do not see that this method is more operative than others, despite the fact it is interesting and deserves to be published. It is the same for the term used geology, I do not see how it is possible to extract the impact on forecasts.

It is also unclear to know to understand in the paper, what is an a priori or an a posteriori information. The way the variability is presented appears to be estimated a posteriori knowing Tf. Maybe I am wrong, but then it means that it is not well explained in the text.

My proposal it to remove the interpretation part and the argument stating that the geomechanics is not the main controlling parameter. But this is obvious from the usual confusion made about creep which is related to a materials, and the landslide failure which is related to a complex body that is controlled by several variables. The creeping does not apply to landslide except in particular cases, this is a general mistake. That is why you can say something about geomechanics, it does not comes from your results, and it can be criticized on fundamental aspects. Then, if you would keep this point, you need to expand the discussion...

The oscillation of the values are interesting, but how do you know that you converge to Tf. In the probability index in the criterion include Tf, which you do not know a priori. Please clarify. You need also to discuss the limitations of the method.

Your work deserves to be published because it is an interesting study, but please clarify the points above and avoid over interpretations. I propose that you present a figure that explain synthetically your process.

Specific comments

- Line 21: define what you means by geomechanics? In the text also.
- Line 46: instead of "is usually" use "can be"

Line 48: you can add reference to the work of Blikra on Aknes rockslide.
Line 49: what do you mean appropriately monitored. In fact, displacements are usually points that often do not represent the global landslide behaviour...

Line 56: 1994 and not 19940

Lines 67-83: references to the works of Dick et al., 2014 (Can Geotech. J., 52, 515–529) and Crosta and Agliardi Can. Geotech. J. 40: 176–191 (2003) and Manconi and Giordan 2015 NHESS.

Line 108: I do not see any probabilistic approach in the paper... There is only stdev of the forecast figure 3.

Line 111-113: this is the heart of the paper. I think you need to develop this and make a small flow chart with graphs to explain you procedure.

Line124- 133: you need to give more information about the assumption of these three equations, which will be helpful for the discussion.

Table 1: for the mechanisms, you must probably refer to a classification Hungr et al., 2015 or Varnes and Cruden (1996).

Figure 2: improve the quality of graphs not simply from excel...

Figure 3: improve quality remove the second box.

Lines 190-197: unclear f Tf must be known?

Line 199: use PI for predictable Index instead of Pi which give the impression of a probability.

Lines 249-251: this is not an argument because with an oscillating process it will always have something very close to the Tf which can be better before collapse.

Line 262-263: as it is presented the predictability index need the knowledge of Tf (see lines 190-197)

---

## Referee Comment (RC3) · Anonymous Referee #3 · 28 Aug 2016

Dear Editor of the NHESSD and authors of the paper nhess-2016-221, here is my review of the paper: The manuscript entitled "Operative and reliable landslide forecasting and influence of geology to predictability" by E. Intrieri and G. Gigli is very interesting and well structured absolutely suitable for the NHSSD. The proposed methodology is innovative and will be appreciated by the landslide prediction researchers. The paper is suitable for publication. Since I m the third reviewer and I have seen the reviews of the two other colleagues I have to say that I agree with most of the issues mentioned by the other Reviewers and I don't need to repeat some of their comments, suggestions and corrections. I just want to repeat that it is not totally correct for the authors to state that "the geomechanics is not the main controlling parameter and that plays an indirect role in landslide predictability". Many more case studies should be investigated to come to this conclusion. I do not see the "involvement" of the geology to the

predictability. Maybe further explanation shuld be provided since it is mentioned in the title of the paper. In my opinion the authors should enrich the discussion about "the limitations of the proposed method". Is it possible to add a map with the locations of the landslides cases used in this study (the events of Table 1). The authors should explain what they mean by the term "artificial landslides". The quality of the diagrams should be improved. A flow diagram of the proposed method would be appreciated by the readers.
* * *

---

## Author Comment (AC1) · 1 Sep 2016

E. Intrieri and G. Gigli

emanuele.intrieri@unifi.it

Reviewer: The authors state the importance of kinematics over geomechanics, based on their interpretation of results. I would suggest that not only does geomechanics play a major role in the kinematics of some of their case studies, but also that predictability of other landslide types not included in the database in this paper are likely controlled by the geomechanics. Clear examples are landslides in sensitive clays and other materials.

Authors: The authors did not mean to diminish the obvious importance of geomechanics to predictability. However, since this point has been unclear for all the reviewers, it is evident that we failed in our explanation. What we mean is that even though geomechanics is unquestionably a key factor, it is sometimes difficult to have a deep

knowledge of the geomechanical features of a landslide, especially in the field and in emergency situations, although some safe assumptions can always been done by observation and a broad knowledge of the area. What it may be known about them is in part thanks to what is derived from displacement data. Like in a black box model, even if the real properties of a phenomenon are not known, we can draw conclusions from the output of those properties (i.e. the kinematics). In this case, importance has been done to kinematics because what is generally measured by monitoring are displacement data and because many other unknown factors (rainfall, ground saturation, earthquakes, anthropic disturbance) are included in the black box together with the geomechanics; this makes it virtually impossible to know in advance what may be the degree of influence of geomechanics alone with respect to other factors, thus leading to focusing on kinematics instead. Moreover, even though geomechanics is a key element, landslide prediction can be carried out with a variety of different geomechanical settings. This explanation can be added in the conclusions, while in the rest of the text every misleading comment that may have reduced the importance of geomechanics will be changed or removed.

R: The authors should also discuss the issue of timely predictability. Methods used to predict landslides that are based on displacement monitoring assume that slope collapse will be preceded by accelerations, sufficiently in advance to make adequate predictions followed by emergency measures. Again, landslides in sensitive clays and other collapsible materials are examples where this assumption might not be valid. Moreover, the recent failure of the Mount Polley Dam (IEEIRP, 2015) suggest that, under certain conditions, undrained responses leading to failure might not provide enough warning time for emergency plans to be in place. It is suggested the authors state such limitations of the methods proposed.

A: Indeed this is an important issue. Our test sites are all cases where timely predictions were possible. However these limitations are not addressable to the method proposed rather than to all the forecasting methods currently available to the scientific community, since some types of landslide still do not allow for a timely prediction. However we agree that this issue could be commented on in the text.

R: The methodology presented addresses the variability of the forecasting methods used. The reliability index, based on this variability, the convergence and non convergence of forecasts; appears to be a measure of data scatter and trend variation, rooted in the behavioural nature of the landslide in its pre-failure stage. To assess the reliability of any forecasting method, the range of forecasts for a number of case studies needs to be compared against observed time of failure. This requires, in my opinion, to subdivide the case dataset in groups of same landslide type, kinematics, materials, triggers, etc., and compare the forecasts with the observed times of failure.

A: The variability, convergence and non convergence of forecasts are already compared with the observed time of failure. In fact, as stated in the text, during the evaluation of the predictability index the time of failure (Tf) is always considered: "1 point: the prediction plot never converges on a single tf (typically tf increases at every new datum available). 2 points: the predictions vary considerably at every new iteration. An average time of failure (t Ǐ_f) can be extracted but with high uncertainty. 3 points: the predictions oscillate around Tf, although with a certain variance. 4 points: the predictions have a low variance although t Ǐ_f is slightly different than Tf. Note that when the variance was low, t Ǐ_f and Tf never differed greatly. 5 points: the prediction plot is clearly centred on Tf therefore the reliability of t Ǐ_f is high." Predictions that oscillate far from Tf are already addressed. Concerning the suggestion of clustering the landslides according to type, kinematics, materials, triggers, etc., we think that, due to the not so large number of landslides, every group would be represented by only few examples and therefore would not be meaningful. However comparisons of behaviours between landslides of the same or different type, kinematics, material, trigger, etc. can easily be done by readers using tables 1 and 2. In any case, as we stated in the text, we already studied such comparisons and did not made interesting findings.

R: 1.- How was brittleness assigned for the cases in Table 1?

A: It was assigned based on information derived from the reference articles. Since it was rarely explicitly stated, we assumed a qualitative level of brittleness based on the type of material, the presence of a reactivated landslide, the weathering and the shape of the displacement curve. Since this leads to approximations we decided to evaluate the brittleness with broad and qualitative definitions.

R: 2.- In Table 1, the event at Vaiont is classifi̧ed as a "Rock Avalanche". This term refers to the material (rock) and its post-failure behaviour. I suggest it should be classifi̧ed following its detachment process, as this is what we are monitoring prior to failure and would give more insight into the role of landslide kinematics vs. predictability.

A: We agree with your observation. Rock slide would be more appropriate. 3.- What are the artifi̧cial landslides? We mean landslides recreated in laboratory. Although from the original paper there is not mention of the dimensions of the artificial slope, a photograph shows that it is big enough not to be called a scale model. We can specify this in the paper.

R: 1.- I suggest the improvement of the excel fi̧gures. fonts are too small, and layout is not technical. The text refers to dashed black and grey lines that appear continuous red and blue in the fi̧gures.

A: Thank you for your observation.

R: 2.- Should the title read "...influence of geology on predictability" rather than "...influence of geology to predictability"?

A: The title has been changed as suggested by all the reviewers. It is now "Of reliable landslide forecasting and factors influencing predictability".

Sincerely, Emanuele Intrieri

---

## Author Comment (AC3) · 1 Sep 2016

E. Intrieri and G. Gigli

emanuele.intrieri@unifi.it

Reviewer: I just want to repeat that it is not totally correct for the authors to state that "the geomechanics is not the main controlling parameter and that plays an indirect role in landslide predictability". Many more case studies should be investigated to come to this conclusion. Authors: see our answers to the previous reviewers.

R: I do not see the "involvement" of the geology to the predictability. Maybe further explanation shuld be provided since it is mentioned in the title of the paper. A: thank you for your observation. The title has been changed into "Of reliable landslide forecasting and factors influencing predictability".

R: In my opinion the authors should enrich the discussion about "the limitations of the proposed method". A: we will add a part in the discussion including all the comments

made by the reviewers concerning this issue.

R: Is it possible to add a map with the locations of the landslides cases used in this study (the events of Table 1). A: unfortunately in the references papers the location is not specified for every landslide therefore the map would be only partial and not meaningful. However in some cases more detailed information can be retrieved from the relative papers.

R: The authors should explain what they mean by the term "artificial landslides". A: We mean landslides recreated in laboratory. Although from the original paper there is not mention of the dimensions of the artificial slope, a photograph shows that it is big enough not to be called a scale model. We can specify this in the paper.

R: The quality of the diagrams should be improved. A: as suggested by reviewer 1, the writings will be increased and the symbols will be coherent with the text.

R: A flow diagram of the proposed method would be appreciated by the readers. A: as suggested also by reviewer 2, this will be added in the discussion.

---

## Author Response (AR1)

**Of reliable landslide forecasting and factors influencing predictability**

**Emanuele Intrieri[1]\*, Giovanni Gigli[1]**

[1] *Department of Earth Sciences, University of Studies of Firenze, via La Pira 4, 50121 Firenze, Italy.*

*\*Corresponding author*

**ABSTRACT**

Forecasting a catastrophic collapse is a key element in landslide risk reduction, but also a very difficult task, owing to the scientific difficulties in predicting a complex natural event and also to the severe social repercussions caused by a false or a missed alarm. A prediction is always affected by a certain error, however when this error can imply evacuations or other severe consequences a high reliability in the forecast is, at least, desirable.

In order to increase the confidence of predictions, a new methodology is here presented. Differently from traditional approaches, it iteratively applies several forecasting methods based on displacement data and, also thanks to an innovative data representation, gives a valuation of how the prediction is reliable. This approach has been employed to back-analyse 15 landslide collapses. By introducing a predictability index, this study also contributes to the understanding of how geology and other factors influence the possibility to forecast a slope failure. The results showed how kinematics, and all the factors influencing it such as geomechanics, rainfall and other external agents, is the key feature that, contrarily to what is generally believed, geomechanics plays an indirect role when concerning in landslide predictability; instead kinematics, and all the factors influencing it, is the key feature.

*Keywords: landslides; forecasting; geomechanics; early warning; time of failure; slope failure*

**INTRODUCTION**

Natural disaster forecasting for early warning purposes is a field of study that drew the media attention after events such as the 26[th] December 2004 tsunami of Sumatra. Predicting landslides, with respect to other natural hazards, is a complex task due to the influence of many factors like geomechanical properties, rainfall, ground saturation, topography, earthquakes and many others. So far, few empirical landslide forecasting methods exist (Azimi et al., 1988; Fukuzono, 1985a; Mufundirwa et al., 2010; Saito, 1969; Voight et al., 1988) and none furnishes a reliability degree about the prediction, making them unsuitable for decision making. In particular when mentioning geomechanics we particularly refer to the study of the behaviour of a landslide concerning its deformation with relation to the applied stress, with particular reference to its post-rupture conditions.

In our research we present an approach to perform probabilistic forecasting of landslides collapse. 
[revised manuscript text omitted]

simply and directly. Instead both depend on the kinematics of the landslide, which in turn
depends on the geological conditions. In the complex relation between geology and kinematics
the aforementioned factors may intervene and asymptotic trends in the inverse velocity plot have
been encountered also for first failure ruptures (as found in some time series of Mount Beni
landslide).
In other words, even though geomechanics is unquestionably a key factor, it is sometimes
difficult to have a deep knowledge of the geomechanical features of a landslide, especially in the
field and in emergency situations, although some safe assumptions can always been done by
observation and a broad knowledge of the area. What it may be known about them is in part

[revised manuscript text omitted]

**Answers to reviewers**

**Reviewer 1**

Reviewer: I appreciate the effort by the authors on pursuing a landslide prediction tool that accounts for the reliability in its predictions. The proposed methodology is based on careful consideration of the work done by others and supported by its implementation on several case studies. This is important work that should be encouraged in landslide research for risk management purposes. I do have some general comments and discussion.
The authors state the importance of kinematics over geomechanics, based on their interpretation of results. I would suggest that not only does geomechanics play a major role in the kinematics of some of their case studies, but also that predictability of other landslide types not included in the database in this paper are likely controlled by the geomechanics. Clear examples are landslides in sensitive clays and other materials prone to collapse.

**Authors: The authors did not mean to diminish the obvious importance of geomechanics to predictability. However, since this point has been unclear for all the reviewers, it is evident that we failed in our explanation.**
**What we mean is that even though geomechanics is unquestionably a key factor, it is sometimes difficult to have a deep knowledge of the geomechanical features of a landslide, especially in the field and in emergency situations, although some safe assumptions can always been done by observation and a broad knowledge of the area. What it may be known about them is in part thanks to what is derived from displacement data. Like in a black box model, even if the real properties of a phenomenon are not known, we can draw conclusions from the output of those properties (i.e. the kinematics). In this case, importance has been done to kinematics because what is generally measured by monitoring are displacement data and because many other unknown factors (rainfall, ground saturation, earthquakes, anthropic disturbance) are included in the black box together with the geomechanics; this makes it virtually impossible to know in advance what may be the degree of influence of geomechanics alone with respect to other factors, thus leading to focusing on kinematics instead. Moreover, even though geomechanics is a key element, landslide prediction can be carried out with a variety of different geomechanical settings. This explanation can be added in the conclusions, while in the rest of the text every misleading comment that may have reduced the importance of geomechanics will be changed or removed.**

R: The authors should also discuss the issue of timely predictability. Methods used to predict landslides that are based on displacement monitoring assume that slope collapse will be preceded by accelerations, sufficiently in advance to make adequate predictions followed by emergency measures. Again, landslides in sensitive clays and other collapsible materials are examples where this assumption might not be valid. Moreover, the recent failure of the Mount Polley Dam (IEEIRP, 2015) suggest that, under certain conditions, undrained responses leading to failure might not provide enough warning time for emergency plans to be in place. It is suggested the authors state such limitations of the methods proposed.

**A: Indeed this is an important issue. Our test sites are all cases where timely predictions were possible. However these limitations are not addressable to the method proposed rather than to all the forecasting methods currently available to the scientific community,**

**since some types of landslide still do not allow for a timely prediction. This issue has been**
**commented on in the text.**
R: The methodology presented addresses the variability of the forecasting methods used. The
reliability index, based on this variability, the convergence and non convergence of forecasts;
appears to be a measure of data scatter and trend variation, rooted in the behavioural nature of
the landslide in its pre-failure stage. To assess the reliability of any forecasting method, the range
of forecasts for a number of case studies needs to be compared against observed time of failure.
This requires, in my opinion, to subdivide the case dataset in groups of same landslide type,
kinematics, materials, triggers, etc., and compare the forecasts with the observed times of failure.
**A: The variability, convergence and non convergence of forecasts are already compared**
**with the observed time of failure. In fact, as stated in the text, during the evaluation of the**
**predictability index the time of failure (Tf) is always considered:**
- **"1 point: the prediction plot never converges on a single $t_f$ (typically $t_f$ increases at**
**every new datum available).**
- **2 points: the predictions vary considerably at every new iteration. An average time**
**of failure ($\bar{t}_f$) can be extracted but with high uncertainty.**
- **3 points: the predictions oscillate around $T_f$, although with a certain variance.**
- **4 points: the predictions have a low variance although $\bar{t}_f$ is slightly different than $T_f$.**
**Note that when the variance was low, $\bar{t}_f$ and $T_f$ never differed greatly.**
- **5 points: the prediction plot is clearly centred on $T_f$ therefore the reliability of $\bar{t}_f$ is**
**high."**

**Predictions that oscillate far from Tf are already addressed.**
**Concerning the suggestion of clustering the landslides according to type, kinematics,**
**materials, triggers, etc., we think that, due to the not so large number of landslides, every**
**group would be represented by only few examples and therefore would not be meaningful.**
**However comparisons of behaviours between landslides of the same or different type,**
**kinematics, material, trigger, etc. can easily be done by readers using tables 1 and 2. In any**
**case, as we stated in the text, we already studied such comparisons and did not make**
**interesting findings.**
R: For particular comments:
1.- How was brittleness assigned for the cases in Table 1?
**A: It was assigned based on information derived from the reference articles. Since it was**
**rarely explicitly stated, we assumed a qualitative level of brittleness based on the type of**
**material, the presence of a reactivated landslide, the weathering and the shape of the**
**displacement curve. Since this leads to approximations we decided to evaluate the**
**brittleness with broad and qualitative definitions. This is now specified in the text.**
R: 2.- In Table 1, the event at Vaiont is classified as a "Rock Avalanche". This term refers to the
material (rock) and its post-failure behaviour. I suggest it should be classified following its
detachment process, as this is what we are monitoring prior to failure and would give more
insight into the role of landslide kinematics vs. predictability.
**A: We agree with your observation. Rock slide is more appropriate.**
R: 3.- What are the artificial landslides?

**A: We mean landslides recreated in laboratory. Although from the original paper there is not mention of the dimensions of the artificial slope, a photograph shows that it is big enough not to be called a scale model. We specified this in the paper.**

R: For editorial comments:

1.- I suggest the improvement of the excel figures. fonts are too small, and layout is not technical. The text refers to dashed black and grey lines that appear continuous red and blue in the figures.

**A: Thank you for your observation. The fonts have been increased. The layout has been changed. Now the symbols are coherent with the text.**

R: 2.- Should the title read "...influence of geology on predictability" rather than "...influence of geology to predictability"?

**A: The title has been changed as suggested by all the reviewers. It is now "Of reliable landslide forecasting and factors influencing predictability".**

R: References:

Independent Expert Engineering Investigation and Review Panel (IEEIRP) (2015) Report on Mount Polley Tailings Storage Facility Breach. Province of British Columbia.
https://www.mountpolleyreviewpanel.ca

**A: Added.**

**Reviewer 2**

Reviewer: Dear Editor, Please find here below my review of the paper nhess-2016-221: Operative and reliable landslide forecasting and influence of geology to predictability By Emanuele Intrieri, Giovanni Gigli This paper is related to new ways of forecasting the time of failure of landslides. It is based on the displacements interpretation by three time to failure existing approaches. The used of the variability of the three methods is proposed to assess the time of failure. The method is applied to several case study. In addition, more general consideration are made about the processes involved.

General comments

The method presented is innovative and interesting, but it seems that too much conclusions are from this research. First the title, is probably to pretentious, I do not see that this method is more operative than others, despite the fact it is interesting and deserves to be published. It is the same for the term used geology, I do not see how it is possible to extract the impact on forecasts.

**Authors: The title has been changed as suggested by all the reviewers. It is now "Of reliable landslide forecasting and factors influencing predictability".**

R: It is also unclear to know to understand in the paper, what is an a priori or an a posteriori information. The way the variability is presented appears to be estimated a posteriori knowing Tf. Maybe I am wrong, but then it means that it is not well explained in the text.

**A: All the case studies are from past landslides that have already failed. Therefore the time of failures are all a posteriori known. In fact, as explained in the method section, the real a posteriori know time of failure is indicated with Tf, while the prediction with tf. This has been clarified in the text.**

R: My proposal it to remove the interpretation part and the argument stating that the geomechanics is not the main controlling parameter. But this is obvious from the usual confusion made about creep which is related to a materials, and the landslide failure which is related to a complex body that is controlled by several variables. The creeping does not apply to landslide
except in particular cases, this is a general mistake. That is why you can say something about
geomechanics, it does not comes from your results, and it can be criticized on fundamental
aspects. Then, if you would keep this point, you need to expand the discussion.
**A: As stated concerning a similar comment of Reviewer 1, the authors did not mean to**
**diminish the obvious importance of geomechanics to predictability. However, since this**
**point has been unclear for all the reviewers, it is evident that we failed in our explanation.**
**What we mean is that even though geomechanics is unquestionably a key factor, it is**
**sometimes difficult to have a deep knowledge of the geomechanical features of a landslide,**
**especially in the field and in emergency situations, although some safe assumptions can**
**always been done by observation and a broad knowledge of the area. What it may be**
**known about them is in part thanks to what is derived from displacement data. Like in a**
**black box model, even if the real properties of a phenomenon are not known, we can draw**
**conclusions from the output of those properties (i.e. the kinematics). In this case,**
**importance has been done to kinematics because what is generally measured by monitoring**
**are displacement data and because many other unknown factors (rainfall, ground**
**saturation, earthquakes, anthropic disturbance) are included in the black box together**
**with the geomechanics; this makes it virtually impossible to know in advance what may be**
**the degree of influence of geomechanics alone with respect to other factors, thus leading to**
**focusing on kinematics instead. Moreover, even though geomechanics is a key element,**
**landslide prediction can be carried out with a variety of different geomechanical settings.**
**This has been clarified in the text, while in the rest of the text every misleading comment**
**that may have reduced the importance of geomechanics have been changed or removed.**
R: The oscillation of the values are interesting, but how do you know that you converge to Tf. In
the probability index in the criterion include Tf, which you do not know a priori. Please clarify.
You need also to discuss the limitations of the method. Your work deserves to be published
because it is an interesting study, but please clarify the points above and avoid over
interpretations. I propose that you present a figure that explain synthetically your process.
**A: the predictability index in fact can only be estimated after the collapse. It has been**
**introduced here as a means to evaluate the performance of the different forecasting**
**methods with different case studies and to allow us to draw conclusions. This has been**
**clarified in the text.**
**Thank you for your suggestion of adding a figure to show the process. It has been added.**
Specific comments
R: Line 21: define what you means by geomechanics? In the text also.
**A: we mean the study of the behaviour of a landslide concerning its deformation with**
**relation to the applied stress, with particular reference to its post-rupture conditions. We**
**are interested in geomechanics especially concerning the issue relative to ductility and**
**brittleness. Now we explained in the text.**
R: Line 46: instead of "is usually" use "can be"
Line 48: you can add reference to the work of Blikra on Aknes rock slide
**A: all have been corrected in the text.**
Line 49: what do you mean appropriately monitored. In fact, displacements are usually points
that often do not represent the global landslide behaviour: : :

**A: Exactly. Moreover monitoring may be carried out for short periods not encompassing**
**the final pre-failure stages, or may have been carried out with too low temporal frequency**
**that do not allow to follow the displacement trend. This has been now explained in the text.**
R: Line 56: 1994 and not 19940
Lines 67-83: references to the works of Dick et al., 2014 (Can Geotech. J., 52, 515–529) and
Crosta and Agliardi Can. Geotech. J. 40: 176–191 (2003) and Manconi and Giordan 2015
NHESS.
**A: these has been changed in the text.**
R: Line 108: I do not see any probabilistic approach in the paper: : : There is only stdev of the
forecast figure 3.
**A: the standard deviation would not be possible with a deterministic approach which is the**
**standard way of applying these forecasting methods, that is every method gives a single**
**prediction. At most more predictions can be made in the future but usually only one (the**
**most recent) is used. With our approach we show not only that the most recent prediction**
**is not necessarily the most accurate, but also that the iteration of the forecasting methods**
**(that is the probabilistic approach) enables to have a standard deviation, that is basically a**
**confidence and a probability distribution.**
R: Line 111-113: this is the heart of the paper. I think you need to develop this and make a small
flow chart with graphs to explain you procedure.
**A: thank you for the suggestion. The figure has been added.**
R: Line124- 133: you need to give more information about the assumption of these three
equations, which will be helpful for the discussion.
**A: as stated in the text, the assumptions of these equations are the presence of the tertiary**
**creep and the absence of external influencing factors such as rainfall (as stated by Voight**
**1988, 1989) although we showed that even in the presence of external factors reliable**
**predictions may still be made. More details can be found in the referenced papers.**
R: Table 1: for the mechanisms, you must probably refer to a classification Hungr et al., 2015 or
Varnes and Cruden (1996).
Figure 2: improve the quality of graphs not simply from excel: : :
Figure 3: improve quality remove the second box.
**A: these have been changed in the text.**
R: Lines 190-197: unclear f Tf must be known?
**A: Yes. See one of our previous comments.**
R: Line 199: use PI for predictable Index instead of Pi which give the impression of a
probability.
**A: Agreed.**
R: Lines 249-251: this is not an argument because with an oscillating process it will always have
something very close to the Tf which can be better before collapse.
**A: this conclusion seems obvious only after that we have demonstrated that predictions**
**often oscillate around the actual time of failure. On the other hand, Rose and Hungr state**
**that only more recent forecasts should be considered, without acknowledging the whole**
**trend.  This is one of the main differences between a probabilistic and a deterministic**
**approach.**
R: Line 262-263: as it is presented the predictability index need the knowledge of Tf (see lines
190-197)
**A: Yes, as explained above.**

**Reviewer 3**

Reviewer: Dear Editor of the NHESSD and authors of the paper nhess-2016-221, here is my
review of the paper: The manuscript entitled "Operative and reliable landslide forecasting and
influence of geology to predictability" by E. Intrieri and G. Gigli is very interesting and well
structured absolutely suitable for the NHSSD. The proposed methodology is innovative and will
be appreciated by the landslide prediction researchers. The paper is suitable for publication.
Since I m the third reviewer and I have seen the reviews of the two other colleagues I have to say
that I agree with most of the issues mentioned by the other Reviewers and I don't need to repeat
some of their comments, suggestions and corrections.
I just want to repeat that it is not totally correct for the authors to state that "the geomechanics is
not the main controlling parameter and that plays an indirect role in landslide predictability".
Many more case studies should be investigated to come to this conclusion.
**Authors: see our answers to the previous reviewers.**

R: I do not see the "involvement" of the geology to the predictability. Maybe further explanation
shuld be provided since it is mentioned in the title of the paper.
**A: thank you for your observation. The title has been changed into "Of reliable landslide**
**forecasting and factors influencing predictability".**

R: In my opinion the authors should enrich the discussion about "the limitations of the proposed
method".
**A: we added a part in the discussion including all the comments made by the reviewers**
**concerning this issue.**

R: Is it possible to add a map with the locations of the landslides cases used in this study (the
events of Table 1).
**A: unfortunately in the references papers the location is not specified for every landslide**
**therefore the map would be only partial and not meaningful. However in some cases more**
**detailed information can be retrieved from the relative papers.**

R: The authors should explain what they mean by the term "artificial landslides".
**A: We mean landslides recreated in laboratory. Although from the original paper there is**
**not mention of the dimensions of the artificial slope, a photograph shows that it is big**
**enough not to be called a scale model. We specified this in the paper.**

R: The quality of the diagrams should be improved.
**A: as suggested by reviewer 1, the writings have been increased and the symbols are now**
**coherent with the text. Graphics also changed.**

R: A flow diagram of the proposed method would be appreciated by the readers.
**A: as suggested also by reviewer 2, this has been added in the discussion.**

---

## Author Response (AR2)

- 1 Of reliable IL and slide forecasting and factors influencing

- predictability Emanuele Intrieri1\*, Giovanni Gigli1 1 Department of Earth Sciences, University of Studies of Firenze, via La Pira 4, 50121 Firenze, 5

- Italy.
- \*Corresponding author

**9 ABSTRACT**

[revised manuscript text omitted]

 iandsnue D
 rainfall
 failure

 \*The references used are numbered as follows: 1: Rose and Hungr, 2007; 2: Zavodni and Broadbent, 1980; 3: Semenza and Melidoro, 1992; 4: Casagli et al., 2009; 5: Gigli et al., 2011; 6: Iovine et al., 2006; 7: Mufundirwa et al., 2010; 8: Saito, 1969; 9: Azimi et al., 1988; 10: Petley et al., 2002; 11: Angeli et al., 1989; 12: Fukuzono, 1985b.
 † The case of Stromboli is not relative to a landslide, rather to a volcanic bulging preceding a vent opening that was forecasted in a similar fashion of a landslide and therefore here included.

224 225

Figure 2. These graphs show how iterating forecasts performed through multiple forecasting methods increases the confidence when estimating the actual time of failurePrediction plots of 226 four different case studies. The dashed line indicates ( $T_{f_2}$  dashed line). The crosses represent 227 228 forecasts performed with S method, the triangles with F method and the diamonds with M 229 method. Note that F forecasts for Avran valley landslide include other less accurate values not 230 showed in the graph as they are out of scale.